# Isolation of Scalarane-Type Sesterterpenoids from the Marine Sponge *Dysidea* sp. and Stereochemical Reassignment of 12-*epi*-Phyllactone D/E

**DOI:** 10.3390/md19110627

**Published:** 2021-11-09

**Authors:** A-Young Shin, Arang Son, Changhoon Choi, Jihoon Lee

**Affiliations:** 1Korea Institute of Ocean Science & Technology (KIOST), Busan 49111, Korea; dkdud1624@kiost.ac.kr; 2Department of Marine Biotechnology, University of Science & Technology, Daejeon 34113, Korea; 3Department of Radiation Oncology, Samsung Medical Center, Seoul 06351, Korea; arang.son@sbri.co.kr (A.S.); changhoon1.choi@samsung.com (C.C.)

**Keywords:** *Dysidea*, sesterterpenoid, scalarane, marine sponge, marine natural product, anticancer activity, stereochemistry reassignment

## Abstract

The chemical investigation of the marine sponge *D**ysidea* sp., which was collected from Bohol province in the Philippines, resulted in the identification of 15 new scalarane-type sesterterpenoids (**1**–**1****4**, **1****6**), together with 15 known compounds. The chemical structures of the new compounds were elucidated based on NMR spectroscopy and HRMS. The structure of 12-*epi*-phyllactone D/E (**15**) isolated during this study was originally identified in 2007. However, careful inspection of our experimental ^13^C NMR spectrum revealed considerable discrepancies with the reported data at C-9, C-12, C-14, and C-23, leading to the correction of the reported compound to the C-12 epimer of **15**, phyllactone D/E. The biological properties of compounds **1**–**16** were evaluated using the MDA-MB-231 cancer cell line. Compound **7**, which bears a pentenone E-ring, exhibits significant cytotoxicity with a GI_50_ value of 4.21 μM.

## 1. Introduction

Sesterterpenoids, such as the ophiobolins, sterigmatocystin, and hippolide A, have attracted a lot of attentions as potent pharmaceutical compounds because of their unique anti-inflammatory activity and cytotoxicity in various cancer cell lines [1]. These compounds are ubiquitous in a broad range of natural sources, from easily accessible terrestrial plants and insects to hard-to-access marine organisms. In marine nature, the scalarane-type scaffolds have emerged as one of the most prevalent structural features of the sesterterpenoids [2]. Since scalarin, a pentacyclic scalarane, was first isolated from the marine sponge *Cacospongia scalaris* in 1972 [3], a number of scalarane-type sesterterpenoids has been isolated from *Dysidea* sp. [4,5], *Phyllospongia* sp. [6,7,8,9,10,11], *Strepsichordaia* sp. [12], *Cateriospongia* sp. [13,14], *Smenospongia* sp. [15], and *H**yrtios* sp. [16], belonging to the order Dictyoceratida [17].

This family of scalarane derivatives is featured with a trans-fused 6/6/6/6 ring system and can be further categorized into three structural subgroups, namely scalarane, homoscalarane, and bishomoscalarane, based on the presence of single carbon substituents at C-20 and/or C-24 (Figure 1). Among them, bishomoscalarane exhibits an exceptionally broad range of diversity in the carbon framework, arising from two distinctive sites: C-20 and C-24/C-25 (Figure 2). Therefore, cyclopropane or alcohol/esters are frequently found at C-20 adjacent to the A ring [12]. The oxidation of C-24 and C-25 results in the formation of an extra E ring in the form of a lactone or cyclopentenone [18]; 24-oxo-25-nor-bishomoscalarane has also been identified as another feature of the D ring [8]. In addition, oxidation of the backbone usually occurs at C-3 [6], C-12 [19], and C-16 [12] to produce hydroxyl or ester substituents. A large group of bishomoscalarane derivatives found in nature is considered to be the outcome of these variations occurring in combinations.

The marine sponge *Dysidea* sp. is known to be a rich source of scalaranes, which exhibits useful pharmacological properties, such as anticancer and antimicrobial activities [17,20,21,22,23,24]. In the course of our studies on bioactive natural products from marine organisms, we inspected the chemical components of *Dysidea sp.* collected from the Bohol province in the Philippines. As a result, we identified 15 new scalarane derivatives, including one scalarane, four 20,24-bishomo-25-norscalaranes, and 10 bishomoscalaranes (Figure 3), along with 14 known compounds (Appendix A, Appendix A). In this report, we disclose the structural assignment of the new scalarane sesterterpenoids and their pharmacological properties as anti-cancer agents. In addition, the C-12 configuration of compound **15**, which was assigned by Li in 2007 [11], was reinvestigated because of the significant differences observed between the reported and experimental ^13^C chemical shifts at C-9, C-12, C-14, and C-23.

## 2. Results and Discussion

### 2.1. Structure Elucidation

Compound **1** was isolated as a colorless oil, and its molecular formula was determined to be C_31_H_48_O_5_ using HRESIMS (*m*/*z* [M + Na]^+^ 523.3382, calcd 523.3394), corresponding to eight degrees of unsaturation (DOU). The ^1^H NMR spectrum of **1** exhibited three singlet methyl groups at *δ*_H_ 0.77, 0.82, and 1.04; three doublet methyl groups at *δ*_H_ 1.07, 1.23, and 1.37; and three oxymethines at *δ*_H_ 4.03, 4.19, and 5.41. Furthermore, unique upfield signals at *δ*_H_ 0.57 and −0.49 indicated the presence of a cyclopropane. Analysis of the ^13^C NMR and HSQC spectra revealed the presence of two ester carbons (*δ*_C_ 174.8 and 171.7), three oxymethine carbons (*δ*_C_ 80.4, 74.8, 64.5), 10 methylene carbons, six methine carbons, and six methyl groups. The HMBC data showed notable correlations between the singlet methyl groups and methines from *δ*_H_ 0.77 to *δ*_C_ 51.4/50.3, *δ*_H_ 0.82 to *δ*_C_ 54.3/51.4, and *δ*_H_ 1.04 to *δ*_C_ 54.3, which are known as characteristic correlations occurring from the ring junctions of scalarane-type 6/6/6/6 fused-cyclic systems (Figure 4). Additional HMBC correlations from the doublet methyl group at *δ*_H_ 1.37 to *δ*_C_ 80.4/44.9 and from the methine at *δ*_H_ 2.34 to *δ*_C_ 174.8/44.9 suggested the existence of a *γ*-valerolactone moiety. Therefore, our preliminary findings led to the hypothesis that compound **1** possessed a honulactone A-like scaffold (B+D type shown in Figure 2) [12].

While the △^17,18^-olefin in honulactones is considered one of the structural features that forms the unsaturated lactone E-ring, the initially identified *γ*-valerolactone and DOU suggest the possibility of a saturated terminal lactone in compound **1**. This speculation was confirmed by the ^1^H-^1^H COSY cross peak observed for H_2_-15–H_2_-16–H-17–H-18, as well as HMBC correlations from CH_3_-23 (*δ*_H_ 1.04) to C-18 (*δ*_C_ 52.5) and from H-18 (*δ*_H_ 2.34) to C-13 (*δ*_C_ 38.7). In addition, the cyclopropane moiety inferred from the ^1^H NMR data was positioned at C-4 based on the HMBC correlations from H_2_-19 (*δ*_H_ 0.57, and –0.49) to C-3 (*δ*_C_ 33.2)/C-5 (*δ*_C_ 50.3) and from CH_3_-27 (*δ*_H_ 1.07) to C-4 (*δ*_C_ 22.7), and the spin system for CH_3_-27–H-20 (*δ*_H_ 0.72)–H_2_-19 (*δ*_H_ 0.57, −0.49) in the ^1^H-^1^H COSY spectrum. Interpretation of the remaining HMBC correlations from CH_3_-4′ (*δ*_H_ 1.23) to C-2′ (*δ*_C_ 43.3)/C-3′ (*δ*_C_ 64.5), H_2_-2′ (*δ*_H_ 2.49/2.42) to C-1′ (*δ*_C_ 171.7)/C-3′, and H-12 (*δ*_H_ 5.41) to C-1′ elucidated the 3-hydroxyl butanoate group at C-12.

The trans-fused cyclic scaffold in **1** was determined from the NOESY cross peaks observed between H-11*β* (*δ*_H_ 1.71) and CH_3_-21 (*δ*_H_ 0.82)/CH_3_-22 (*δ*_H_ 0.77), and CH_3_-23 and CH_3_-21/H-17 (*δ*_H_ 1.86) (Figure 5). The NOESY correlations between H-12 and CH_3_-23, and H-18 and H-14 (*δ*_H_ 1.23)/H-24 (*δ*_H_ 4.03) suggested the *β*-orientations of H-12 and CH_3_-26, respectively. Moreover, the 20*S** configuration of CH_3_-27 was determined based on the NOESY signals observed between H-19*_cis_* (δ_H_ –0.49) and H-3*β* (*δ*_H_ 1.24)/CH_3_-27, and H-19*_trans_* (*δ*_H_ 0.57) and H_2_-6.

Compound **2** was isolated as a colorless oil, and its molecular formula was determined to be C_31_H_46_O_6_ by HRESIMS (*m*/*z* [M + Na]^+^ 537.3167, calcd 537.3187), corresponding to nine degrees of unsaturation. Analysis of the 1D and 2D NMR spectra obtained for **2** indicated a similar carbon framework to **1**, but the higher oxidation state of the lactone in E ring appeared as a major difference. HMBC correlations from CH_3_-23 (*δ*_H_ 1.22) to C-18 (*δ*_C_ 133.7) and from CH_3_-26 (*δ*_H_ 1.56) to C-17 (*δ*_C_ 162.9) revealed an *α*,*β*-unsaturated lactone in the E ring, which was responsible for the one degree higher DOU than that of **1**. In addition, the ^13^C chemical shift of C-24 (*δ*_C_ 104.4) was characteristic of a ketal carbon atom, of which the position was confirmed by HMBC correlations from CH_3_-26 to C-24. The *β*-configuration of OH-24 was determined by the NOESY correlation observed between H-16*α* (*δ*_H_ 2.28) and CH_3_-26 (Appendix A, Appendix A).

Compound **3** was isolated as a mixture of two inseparable epimers. The molecular formula of **3** was deduced to be C_31_H_44_O_6_ by HRESIMS (*m*/*z* [M + Na]^+^ 535.3011, calcd 535.3030), corresponding to 10 degrees of unsaturation. An initial inspection of the ^13^C NMR spectrum revealed that most of the peaks were split into a doublet-like shape, indicating a 1:1 mixture of diastereomers. The 1D and 2D NMR spectra obtained for compound **3** exhibited most of the structural features of **2**, except for one more disubstituted olefin observed at *δ*_H_ (6.38/6.37)/*δ*_C_ (138.84/138.80) and *δ*_H_ (6.29/6.25)/*δ*_C_ (118.6/118.4). The location of the double bond was determined to be △^1^^5,16^ using the consecutive ^1^H-^1^H COSY correlations observed for H-14 (*δ*_H_ 2.69/2.62)–H-15 (*δ*_H_ 6.38/6.37)–H-16 (*δ*_H_ 6.29/6.25). The splittings observed in the ^13^C NMR spectrum were most prominent at CH_3_-26 (Δ*δ*_C_ 1.13 ppm), informing a mixture of C-24 epimers. This phenomenon has often been observed in the case of 24-homoscalaranes, which possess both an △^1^^5,16^-olefin and 24-hydroxy pentenolide E-ring [25,26]. Since the △^1^^5,16^-olefin increases the planarity of the D-ring and renders the C-24 stereocenter more isolated, the 24*R** and 24*S** diastereomers exhibit almost identical spectroscopic and chromatographic behaviors to give an inseparable mixture.

Compound **4** was isolated as an inseparable mixture and its molecular formula was determined to be C_32_H_46_O_6_ by HRESIMS (*m*/*z* [M + Na]^+^ 549.3163, calcd 549.3187), indicating 10 degrees of unsaturation. The NMR spectra of **4** were only discriminated from those of **3** by the extra methylene group observed at *δ*_H_ 1.50 and *δ*_C_ 29.5/29.4, which was also supported by the mass difference of +14. The extra methylene group was observed in the ester side chain located at C-12, which formed a 3-hydroxypentanoate moiety, as supported by the spin system for H_2_-2′ (*δ*_H_ 2.35)–H-3′ (*δ*_H_ 3.90/3.86)–H_2_-4′ (*δ*_H_ 1.50)–CH_3_-5′ (*δ*_H_ 0.95) in the ^1^H-^1^H COSY spectrum.

Compound **5** was isolated as a colorless oil. Its molecular formula was determined to be C_32_H_48_O_6_ by HRESIMS (*m*/*z* [M + Na]^+^ 551.3310, calcd 551.3343), corresponding to nine degrees of unsaturation. Our initial analysis of the ^1^H NMR spectrum obtained for compound **5** indicated the presence of the scalarane-type scaffold: five singlet methyl groups at *δ*_H_ 0.80, 0.84, 1.06, 2.02, and 2.22; two doublet methyl groups at *δ*_H_ 1.08 and 1.25; three oxymethines at *δ*_H_ 4.19, 5.11, and 5.76; unique cyclopropane signals at *δ*_H_ 0.59 and –0.49; and one singlet olefin at *δ*_H_ 6.72. The ^13^C NMR and HSQC spectra showed one ketone (*δ*_C_ 197.7), two ester carbons (*δ*_C_ 172.0, 170.2), one trisubstituted olefin carbon (*δ*_C_ 153.2, 135.1), three oxymethine carbons (*δ*_C_ 76.8, 65.3, 64.6), eight methylenes, four methines, and seven methyl groups. In addition, the HMBC correlation from the singlet methyl at *δ*_H_ 2.22 to *δ*_C_ 197.7 suggested the presence of a methyl ketone moiety, instead of the lactone E-ring observed in compounds **1**–**4**, leading to the conclusion that **5** had a B+F type skeleton, as shown in Figure 2.

Detailed interpretation of the combined spectral data of **5** revealed that the features related to the A-B-C ring system were identical to those of **1**–**4**. As anticipated, the methyl ketone was positioned at C-17 to form an unsaturated ketone in the D ring on the basis of HMBC correlations between CH_3_-26 (*δ*_H_ 2.22) and C-17 (*δ*_C_ 135.1), and H-18 (*δ*_H_ 6.72) and C-17/C-24 (*δ*_C_ 197.7) (Figure 4). Moreover, the ^1^H-^1^H COSY cross peak for H-14 (*δ*_H_ 1.76)–H_2_-15 (*δ*_H_ 1.89, 1.61)–H-16 (*δ*_H_ 5.76) and HMBC correlations from *δ*_H_ 2.02 (C*H*_3_CO_2_–) to *δ*_C_ 170.2 (CH_3_*C*O_2_–) and from H-16 to *δ*_C_ 170.2 positioned an acetate substituent at C-16, of which the relative configuration was assigned to be *α*-orientation based on the small coupling constants between H_2_-15 and H-16 (dd, *J*_H-15–H-16_ = 4.3, 1.6 Hz).

Compound **6** was isolated as an amorphous solid. Its molecular formula was determined to be C_30_H_46_O_5_ by HRESIMS (*m*/*z* [M + Na]^+^ 509.3215, calcd 509.3237), corresponding to eight degrees of unsaturation. The ^1^H and ^13^C NMR spectra obtained for compound **6** were almost identical to those of **5**. However, the absence of one ester carbon and the singlet methyl group at *δ*_H_ 2.02 suggested deacetylation from **5**, which was further supported by an upfield shift of H-16 (*δ*_H_ 4.62). The relative configuration of OH-16 was assigned as *β*-orientation based on the large coupling constant observed between H-16 and H-15*β* (dd, *J*_H-15–H-16_ = 9.6, 5.1 Hz).

Compound **7** was isolated as a yellow oil. Its molecular formula was determined to be C_31_H_44_O_4_ by HRESIMS (*m*/*z* [M + Na]^+^ 503.3113, calcd 503.3132), corresponding to 10 degrees of unsaturation. Preliminary analysis of the ^1^H and ^13^C NMR data revealed that the scalarane-type scaffold had a cyclopropane substituent on the A ring. Interpretation of the ^13^C NMR and HSQC spectra exhibited the sp^2^ carbons in the enone systems: three sp^2^ methines at *δ*_H_ 7.38/*δ*_C_ 157.5, *δ*_H_ 6.34/*δ*_C_ 137.4, and *δ*_H_ 6.64/*δ*_C_ 130.4; and one trisubstituted sp^2^ carbon atom at *δ*_C_ 136.4. Therefore, HMBC correlations observed from H-25 (*δ*_H_ 7.38) to C-17 (*δ*_C_ 136.4)/C-18 (*δ*_C_ 49.3)/C-24 (*δ*_C_ 195.9), H-26 (*δ*_H_ 6.34) to C-17/C-24/C-25 (*δ*_C_ 157.5), and H-18 (*δ*_H_ 3.35) to C-17/C-23 (*δ*_C_ 14.5), as well as the ^1^H-^1^H COSY cross peak for H-18–H-25–H-26, confirmed the presence of a △^25,26^-cyclopenten-24-one subunit for the E-ring and the trisubstituted double bond at △^1^^6,17^ (Figure 4).

Compound **8** was isolated as a yellow oil, and its molecular formula was determined to be C_34_H_52_O_8_ by HRESIMS (*m*/*z* [M + Na]^+^ 611.3541, calcd 611.3554), corresponding to nine degrees of unsaturation. The ^1^H NMR spectrum obtained for compound **8** showed similar patterns to that of **5**. However, the upfield peaks observed for the cyclopropane moiety in **5** were substituted by an oxymethine at *δ*_H_ 5.35, a methyl singlet at *δ*_H_ 1.09, and an acetate at *δ*_H_ 2.03, suggesting the C+F type scaffold shown in Figure 2. Therefore, the connectivity of C-27–C-20–C-4–C-19 was determined using the HMBC correlations observed from CH_3_-19 (*δ*_H_ 0.99) to C-20 (*δ*_C_ 73.2) and from CH_3_-27 (*δ*_H_ 1.09) to C-4 (*δ*_C_ 39.4)/C-20 (Figure 6). In addition, the acetate at *δ*_H_ 2.03 exhibited a HMBC correlation with C-20 to be located at C-20. The relative configuration at C-20 was assigned as 20*R** from the NOESY correlations observed between H-20 (*δ*_H_ 5.35) and H-2*β* (*δ*_H_ 1.47)/CH_3_-22 (*δ*_H_ 0.87), and H-3*β* (*δ*_H_ 1.67) and CH_3_-27 (Figure 7). Similarly, the configuration of the acetate group at C-16 was assigned as *α*-orientation based on the small coupling constant observed for H-16 (dd, *J*_H-15–H-16_ = 4.3, 1.6 Hz).

Compound **9** was isolated as a colorless oil, and its molecular formula was determined to be C_30_H_48_O_6_ by HRESIMS (*m*/*z* [M + NH_4_]^+^ 522.3810, calcd 522.3789) corresponding to seven degrees of unsaturation. Analysis of the 1D and 2D NMR data provided almost identical features to those of **8** to determine the carbon skeleton of compound **9**. In this case, only one ester carbon atom (*δ*_C_ 172.2) was observed in the ^13^C NMR spectrum, and the acetate groups shown in the ^1^H NMR spectrum of **8** disappeared. This information indicated that compound **9** was the deacetylation product of **8**. Accordingly, the upfield shifts of H-20 (*δ*_H_ 4.32) and H-16 (*δ*_H_ 4.55) were the major differences, compared to compound **8**.

Compound **10** was isolated as a yellow oil, and its molecular formula was determined to be C_33_H_50_O_8_ by HRESIMS (*m*/*z* [M + Na]^+^ 597.3404, calcd 597.3398), corresponding to nine degrees of unsaturation. Preliminary inspection of the ^13^C NMR and HSQC data of **10** identified four singlet methyl groups (*δ*_H_ 0.87/*δ*_C_ 16.6, *δ*_H_ 0.87/*δ*_C_ 16.8, *δ*_H_ 0.96/*δ*_C_ 23.3, *δ*_H_ 1.13/*δ*_C_ 19.8), three doublet methyl groups (*δ*_H_ 1.07/*δ*_C_ 16.0_,_
*δ*_H_ 1.18/*δ*_C_ 22.5, *δ*_H_ 1.39/*δ*_C_ 18.2), and one acetate group (*δ*_H_ 2.03/*δ*_C_ 22.0), indicating a honulactone C-like scaffold (C+D type shown in Figure 2) [12]. A detailed analysis of the ^1^H NMR spectrum identified an oxymethine group at *δ*_H_ 4.44 as a major difference from honulactone C. The location of the oxymethine was determined to be C-16, as indicated by the HMBC correlations from H-16 (*δ*_H_ 4.44) to C-17 (*δ*_C_ 162.1)/C-18 (*δ*_C_ 135.6) and ^1^H-^1^H COSY cross peak for H_2_-15 (*δ*_H_ 1.91, 1.84)–H-16 (Figure 6). The configuration of the OH-16 group was assigned as *α*-orientation based on the small coupling constant observed for H-16 (dd, *J*_H-15–H-16_ = 4.7, 1.4 Hz), and compound **10** was named as 16*α*-hydroxyhonulactone C [12].

Compound **11** was isolated as a yellow oil. Its molecular formula was determined as C_33_H_50_O_8_ by HRESIMS (*m*/*z* [M + Na]^+^ 597.3396, calcd 597.3398), corresponding to nine degrees of unsaturation. The ^1^H and ^13^C NMR data of **11** were almost identical to those of **10**, but a ketal moiety (*δ*_C_ 104.4) was observed instead of one doublet methyl group and two oxymethines in compound **10**. As shown in compounds **2**–**4**, the hemiketal functionality in the scalarane-type scaffold usually occurs at C-24 in the E-ring, which was also applicable in this case, as indicated by the HMBC correlations from CH_3_-26 (*δ*_H_ 1.56) to C-17 (*δ*_C_ 162.9)/C-24 (*δ*_C_ 104.4). The *α*-orientation of the hydroxyl group at C-24 was determined by the NOESY correlation between H-16*β* (*δ*_H_ 2.33) and CH_3_-26. Thus, compound **11** was named 24*α*-hydroxyhonulactone C [12].

Compound **12** was isolated as an inseparable mixture. Its molecular formula was determined as C_33_H_48_O_8_ by HRESIMS (*m*/*z* [M + Na]^+^ 595.3241, calcd 595.3241), corresponding to 10 degrees of unsaturation. Compared to **11**, two more sp^2^ methines at *δ*_C_ 138.9/138.7 and *δ*_H_ 6.38, and *δ*_C_ 118.44/118.35 and *δ*_H_ 6.28/6.26 were observed in the ^13^C NMR and HSQC spectra, indicating the presence of a disubstituted double bond. These sp^2^ protons were involved in a spin system for H-14 (*δ*_H_ 2.66/2.62)–H-15 (*δ*_H_ 6.38)–H-16 (*δ*_H_ 6.28/6.26) in the ^1^H-^1^H COSY spectrum and used to confirm the presence of the △^1^^5,16^-olefin, which was further supported by HMBC correlations from H-15 to C-13 (*δ*_C_ 40.1/40.0)/C-14 (*δ*_C_ 53.96/53.90)/C-17 (*δ*_C_ 157.3) and from H-16 to C-14/C-18 (*δ*_C_ 130.9). As discussed in the cases of **3** and **4**, the presence of the olefin at △^1^^5,16^ and the hemiketal at C-24 rendered compound **12** an inseparable mixture of C-24 epimers.

Compound **13** was isolated as an amorphous solid. Its molecular formula was determined as C_31_H_48_O_6_ by HRESIMS (*m*/*z* [M + Na]^+^ 539.3325, calcd 539.3343), corresponding to eight degrees of unsaturation. Inspection of the ^1^H NMR spectrum of **13** revealed most of the structural features of the bishomoscalarane-type skeletons. Precise analysis of the ^13^C NMR and HSQC data revealed the presence of a triplet methyl group (*δ*_H_ 0.67/*δ*_C_ 8.80) and ketal carbon (*δ*_C_ 104.4), suggesting the A+D type skeleton shown in Figure 2. While most of the spectral data of **13** were identical to phyllofolactone H, the ketal carbon indicated the oxidation of C-24 to give a 24-hydroxy pentenolide E ring. This insight can be confirmed by the HMBC correlation from CH_3_-26 (*δ*_H_ 1.48) to C-17 (*δ*_C_ 163.0)/C-24 (*δ*_C_ 104.4). The configuration of OH-24 was determined to be *α*-orientation by the NOESY correlation between H-16*β* (*δ*_H_ 2.33) and CH_3_-26. Thus, compound **13** was named 24*α*-hydroxyphyllofolactone H [19].

Compound **14** was isolated as an inseparable mixture. Its molecular formula was determined as C_31_H_46_O_6_ by HRESIMS (*m*/*z* [M + Na]^+^ 537.3175, calcd 537.3187), corresponding to nine degrees of unsaturation. Similar to compound **3**, the ^13^C NMR spectrum of **14** showed a 1:1 splitting pattern corresponding to a mixture of two diastereomers. The distinctive spectral features of **14**, differentiated from **13**, were observed as the two sp^2^ methines at *δ*_H_ 6.40/6.38 and 6.28/6.27, suggesting an unsaturated derivative of **13**. The methines belonged in the ^1^H-^1^H COSY correlation for H-14 (*δ*_H_ 2.68/2.62)–H-15 (*δ*_H_ 6.40/6.38)–H-16 (*δ*_H_ 6.28/6.27) to identify the olefin at C-15 (Figure 6). In addition, **14** was determined to be a mixture of C-24 epimers, considering the largest splitting observed at CH_3_-26 (Δ*δ*_C_ 1.13 ppm).

Compound **15** was isolated as an inseparable mixture. Its molecular formula was determined to be C_32_H_48_O_6_ by HRESIMS (*m*/*z* [M + Na]^+^ 551.3366, calcd 551.3343), corresponding to nine degrees of unsaturation. The MS data indicated an additional methylene relative to **14**, which was further supported by the change observed in the coupling pattern of the terminal methyl group of the side chain at C-12 from a doublet to triplet. The ^13^C NMR and HSQC data identified the methylene group at *δ*_H_ 1.51/1.25 and *δ*_C_ 29.5/29.4, which were involved in the spin system for H_2_-2′–H-3′–H_2_-4′–CH_3_-5′ in the ^1^H-^1^H COSY spectrum to confirm the presence of the 3-hydroxypentanoate side chain. The orientation of the ester at C-12 was assigned as *α* by the NOESY signal between H-12 (*δ*_H_ 5.55/5.49) and CH_3_-23 (*δ*_H_ 1.06/1.05), as well as the small coupling constant observed for H-12 (dd, *J* = 2.3, 1.8 Hz), to identify 12-*epi*-phyllactone D/E.

Interestingly, the identified structure was previously isolated as a mixture of C-24 epimers by Li et al. in 2007 [11], but our experimental ^13^C NMR data showed some discrepancies with the previously reported data at C-9 (Δ 4.28 ppm), C-11 (Δ 2.7 ppm), C-12 (Δ 2.08 ppm), C-14 (Δ 4.46 ppm), and C-23 (Δ 4.35 ppm) (Figure 8a). In addition, another identification of 12-*epi*-phyllactone D/E was reported by Andersen et al. in 2009 [13]. Although they acquired almost identical experimental NMR data with ours rather than those reported by Li, the isolated compound was estimated to be same as Li’s without consideration of the differences in NMR data (Appendix A). Therefore, we investigated the variations in ^13^C chemical shifts depending on the orientation of the substituents at C-12.

Phyllactone D (**17**) and E (**18**), the reported 12*β*-epimers of **15**, were selected for comparison [25]. While C-12 in phyllactones D and E was observed at *δ*_C_ 75.1 and 75.8, respectively, the corresponding chemical shifts of the reported and isolated **15** were observed at *δ*_C_ 75.3 and 73.2/73.1, respectively. The deviations observed for isolated **15** from phyllactone D/E became more obvious at C-9, C-14, and C-23 (Figure 8b). However, the reported chemical shifts for **15** were better aligned with those of phyllactone D/E. Furthermore, the differences in the ^13^C NMR chemical shifts observed between isolated **15** and compounds **3**, **4**, **12**, and **14**, which share an identical substructure for the B-E ring system, showed negligible values (< 0.5 ppm) around the C-ring (Appendix A). Accordingly, isolated **15** is more likely to be the 12*α*-epimer. Even though Li determined the 12*a*-configuration observing the NOESY signal between H-12 and CH_3_-23 and *J*_H-12–H-13_ calculation (3.0, 2.5 Hz), the NMR database suggests that the compound previously reported by Li is presumed to be a mixture of phyllactone D (**17**) and E (**18**).

Compound **16** was isolated as a yellowish oil. Its molecular formula was determined as C_27_H_40_O_5_ by HRESIMS (*m*/*z* [M + Na]^+^ 467.2762, calcd 467.2768), corresponding to eight degrees of unsaturation. The ^1^H NMR spectrum of **16** revealed five singlet methyl groups at *δ*_H_ 0.73, 0.74, 0.79, 0.85, and 0.86; one acetate group at *δ*_H_ 1.95; one oxymethine at *δ*_H_ 4.80; one olefin at *δ*_H_ 7.30; and one aldehyde at *δ*_H_ 9.41. The ^13^C and HSQC NMR spectra showed characteristic peaks for the aldehyde carbon atom at *δ*_C_ 196.4, two carbonyl carbons at *δ*_C_ 169.6 and 169.6, one trisubstituted olefin at *δ*_C_ 145.8 and 124.2, and one oxymethine at *δ*_C_ 76.9. The HMBC correlation between the two methyl groups at *δ*_C_ 33.3 and 21.4 was identified as a characteristic feature of the 4-dimethyl-sesterterpenoid scaffold (Figure 9). The aldehyde at *δ*_H_ 9.41 exhibited a HMBC correlation with C-18 (*δ*_C_ 58.7) to be located at C-25. Additional HMBC correlations from H-18 (*δ*_H_ 3.07) to C-16 (*δ*_C_ 145.8)/C-17 (*δ*_C_ 124.2)/C-24 (*δ*_C_ 169.6), along with the ^1^H-^1^H COSY cross peak for H-14–H_2_-15–H-16, indicated the presence of the acid at C-24 and trisubstituted olefin at C-16. The acetate group (*δ*_C_ 21.3/*δ*_H_ 1.95) was positioned at C-12, as indicated by the HMBC correlation from H-12 (*δ*_H_ 4.80) to C-1′ (*δ*_C_ 169.6) and ^1^H-^1^H COSY cross peak for H_2_-11–H-12. Thus, the planar structure of **16** was found to be the deacetalization product of scalarin (**19**) [3]. The NOESY correlations between CH_3_-23 (*δ*_H_ 0.86) and H-12/H-18 determined the configuration of the C-12 acetate and C-18 formyl groups as *α*.

Whereas scalarin (**19**) exists only in its hemiacetal form, the formation of 18-*epi-***19** or **19** via the acetalization of **16** was not observed. To rationalize the observed difference in reactivity, 18-*epi*-**16** was proposed as a plausible precursor of scalarin, and geometrical optimization of **16** and 18-*epi*-**16** was performed at the B3LYP/6-31G** level of theory. The atomic distance between O-24 to C-25 was calculated to be 3.37 Å for **16** and 2.68 Å for 18-*epi*-**16** (Figure 10). This result suggests that 18-*epi*-**16** can undergo acetalization to form scalarin because the *β*-orientation of C-25 increases its proximity to the acid at C-24. However, the acetalization of the 25*α*-formyl group in **16** will be restricted due to its remoteness to OH-24 to exist as its aldehyde form.

### 2.2. Biological Activity

The cytotoxicity of compounds **1**–**16** against MDA-MB-231 (a human breast cancer cell line) was evaluated to elucidate their potential as anticancer agents. Compounds **1**–**6**, **8**, **11**, and **13**–**15** exhibited moderate cytotoxicity with GI_50_ values ranging from 40 to 72 μM. Compounds **9**, **10**, **12**, and **16** were inactive toward the cancer cell line (Table 1). Among the bishomoscalaranes, the highest anticancer activity was exhibited by compound **7**, which has a relatively rare cyclopentenone E-ring (B+E type scaffold in Figure 2), with a GI_50_ value of 4.2 μM.

The highly diversified structures of the isolated scalaranes provided some information on their structure–activity relationship (SAR). The presence of the △^15,16^-olefin generally had a detrimental effect that reduced the cytotoxicity in the range of 12–30 μM, as shown by the sets of **2** and **3** (B+D type)**, 13** and **14** (C+D type), and **13** and **14** (A+D type). Comparing **3** with **4** and **14** with **15**, the homologation of one methylene group at C-4′ was beneficial toward increasing the activity to ~20 μM. A series of compounds **2**, **11**, and **13**, which only differ at the C-4 substituent, indicated the disadvantageous effect of oxidation at C-20 on the anticancer activity. The negative effect of oxidation at C-20 was also observed in the inactive series of compounds **9**, **10**, and **12**.

## 3. Materials and Methods 

### 3.1. General Experimental Procedures

Specific optical rotations were collected on a Rudolph Research Analytical (Autopol III) polarimeter (Rudolph Research Analytical, Hackettstown, NJ, USA). IR spectra were measured on a JASCO FT/IR-4100 spectrophotometer (JASCO Corporation, Tokyo, Japan). The 1D and 2D NMR spectra were taken in CDCl_3_ using a Bruker 600 MHz spectrometer (Bruker BioSpin GmbH, Rheinstetten, Germany) at 297.1 K. ^1^H NMR spectra were collected after 64–128 scans, and ^13^C NMR spectra were collected at a range of 10,000–15,000 scans depending on the sample concentrations. The mixing time for NOESY experiments was set as 0.3 s. Chemical shifts were reported in parts per million relative to CHCl_3_ residue (*δ*_H_ 7.26, *δ*_C_ 77.1) in CDCl_3_. High resolution mass-spectra were obtained on a Sciex X500R Q-TOF spectrometer (Framingham, MA, USA) equipped with an ESI source. MPLC was performed using the TELEDYNE ISCO CombiFlash Companion with the TELEDYNE ISCO RediSep Normal-phase Silica Flash Column (Teledyne ISCO, Lincoln, NE, USA). HPLC was performed on a PrimeLine Binary pump (Analytical Scientific Instruments, Inc., El Sobrante, CA, USA) utilizing silica columns (YMC-Pack Silica, 250 × 10 mm I.D., or 250 × 4.6 mm I.D., 5 µm; YMC Co. Ltd., Kyoto, Japan), the Shodex RI-101 (Shoko Scientific Co. Ltd., Yokohama, Japan), or the UV-M201.

### 3.2. Biological Material

The marine sponge used in this study was collected in March 2016 from the Bohol province in the Philippines (N 9°43′31.39″ E 124°32′19.86″) at a depth of 15 m using scuba diving. The sponge was directly kept frozen at −20 °C until identified as *Dysidea* sp. and chemically analyzed. A voucher sample (163PIL-267) has been stored at the Marine Biotechnology Research Center, Korea Institute of Ocean Science & Technology (KIOST).

### 3.3. Extraction and Isolation

The lyophilized specimen (wet wt. 1.5 kg) was extracted with MeOH (2.0 L × 3) and CH_2_Cl_2_ (4.0 L × 2) at room temperature. The combined extracts were concentrated under reduced pressure. The dried residue (89.5 g) was partitioned with *n*-butanol (5.0 L) and water (5.0 L). The *n*-butanol layer was concentrated and further partitioned between *n*-hexane (3.0 L) and 15% aqueous methanol (3.0 L). A portion (12.2 g) of the concentrated 15% aqueous methanol fraction (31.7 g) was subjected to flash column chromatography over C18 (YMC Gel ODS-A, 60 Å, 230 mesh (YMC Co, Ltd., Kyoto, Japan)) with a stepwise gradient solvent system (50%, 60%, 70%, 80%, 90%, and 100% MeOH, acetone, and EtOAc).

The 80% MeOH fraction (612.7 mg) was further separated using MPLC on C18 with a gradient solvent system from 70% MeOH to 100% MeOH over 40 minutes to yield 4 fractions. The third subfraction (250.1 mg) was separated using HPLC (eluent 65% MeOH) to yield **8** (3.9 mg, *t*_R_ = 38 min), **9** (2.5 mg, *t*_R_ = 42 min), **10** (2.3 mg, *t*_R_ = 58 min), honulactone C (9.8 mg), and honulactone D (9.0 mg). The fourth subtractions (175.3 mg) was separated using HPLC (eluent 70% MeOH) to yield **11** (1.8 mg, *t*_R_ = 28 min), **12** (1.4 mg, *t*_R_ = 28 min), and honulactone I+J mixture (1.6 mg).

The 100% MeOH fraction (4.22 g) was further separated using MPLC on C18 with a gradient solvent system from 30% MeOH to 100% MeOH over 40 minutes to yield 4 fractions. The second fraction (2.49 g) was directly separated using MPLC on SiO_2_ with a gradient solvent system from 70% HX to 100% EtOAc over 80 minutes to yield 8 subfractions (based on TLC analysis). Scalarin (**19**, 213.0 mg) was recrystallized from the second subfraction (572.8 mg) under the HX-EtOAc solvent conditions. The residue (250.0 mg) of the second subfraction was separated using HPLC (HX/acetone = 7/1) to yield **4** (5.2 mg, *t*_R_ = 54 min), **15** (5.5 mg, *t*_R_ = 48 min), phyllofolactone H (5.7 mg), and phyllofolactone I (11.5 mg). The third subfraction (295.5 mg) was separated using HPLC (HX/acetone = 7/1) to yield **1** (3.4 mg, *t*_R_ = 34 min), **3** (7.0 mg, *t*_R_ = 76 min), **14** (6.0 mg, *t*_R_ = 66 min), **13** (2.3 mg, *t*_R_ = 60 min), **16** (4.5 mg, *t*_R_ = 45 min, honulactone A (21.6 mg), honulactone B (26.2 mg), honulactone E+F mixture (21.4 mg), and phyllofolactone J+K (2.7 mg). The fourth subfraction (380.0 mg) was separated using HPLC (HX/acetone = 5/1) to yield **2** (3.1 mg, *t*_R_ = 36 min), **5** (1.6 mg, *t*_R_ = 31 min), **6** (5.6 mg, *t*_R_ = 32 min), **7** (5.0 mg, *t*_R_ = 30 min), and phyllofenone C (2.3 mg).

### 3.4. Assay

Human breast cancer MDA-MB-231 cells were purchased from the American Type Culture Collection (ATCC, Manassas, VA, USA) and were cultured in Dulbecco’s modified Eagle medium (DMEM) supplemented with 10% fetal bovine serum (FBS, Gibco, Carlsbad, CA, USA), 1 × antibiotic-antimycotic solution (Thermo Fisher Scientific, Waltham, MA, USA), and 25 mM HEPES (Gibco). Cultures were maintained in a humidified atmosphere of 95% air/5% CO_2_ at 37 °C.

Cell viability was determined using a CCK-8 (Cell Counting Kit-8, Dojindo Laboratory, Kumamoto, Japan) assay according to the manufacturer’s instructions. MDA-MB-231 cells were seeded at 5 × 10^3^ cells/well into a 96-well plate and then were treated with various concentrations of compounds **1**–**16**. Following treatment for 48 h, the cells were incubated with the CCK-9 solution, and the absorbance was measured at 450 nm using a SpectraMax i3 microplate reader (Molecular Devices, Sunnyvale, CA, USA). GI_50_ values were calculated from a non-linear regression fit using GraphPad Prism version 9.2.0 (GraphPad Software, La Jolla, CA, USA).

**1**: colorless oil; ${\left[\alpha \right]}_{D}^{20}$ + 20.0 (*c* 0.2, CHCl_3_); IR (ATR) *ν*_max_ 3131, 2954, 2929, 2581, 1770, 1734, 1452, 1381, 1261, 1176, 1027 cm^−1^; ^1^H NMR and ^13^C NMR, see Appendix A; HRESIMS *m*/*z* 523.3382 [M + Na]^+^ (calcd for C_31_H_48_O_5_Na, 523.3394).

**2**: colorless oil; ${\left[\alpha \right]}_{D}^{20}$ + 40.0 (*c* 0.2, CHCl_3_); IR (ATR) *ν*_max_ 3735, 2954, 2925, 2851, 1731, 1689, 1452, 1374, 1278, 1176, 1014 cm^−1^; ^1^H NMR and ^13^C NMR, see Appendix A; HRESIMS *m*/*z* 537.3167 [M + Na]^+^ (calcd for C_31_H_46_O_6_Na, 537.3187).

**3**: colorless oil; ${\left[\alpha \right]}_{D}^{20}$ + 45.0 (*c* 0.2, CHCl_3_); IR (ATR) *ν*_max_ 3727, 2957, 2922, 2865, 2848, 1738, 1657, 1458, 1371, 1286, 1621, 1173, 1031 cm^−1^; ^1^H NMR and ^13^C NMR, see Appendix A; HRESIMS *m*/*z* 535.3011 [M + Na]^+^ (calcd for C_31_H_44_O_6_Na, 535.3030).

**4**: colorless oil; ${\left[\alpha \right]}_{D}^{20}$ + 48.3 (*c* 0.2, CHCl_3_); IR (ATR) *ν*_max_ 3735, 2954, 2922, 2869, 2855, 1731, 1685, 1452, 1374, 1286, 1173, 1021 cm^−1^; ^1^H NMR and ^13^C NMR, see Appendix A; HRESIMS *m*/*z* 549.3163 [M + Na]^+^ (calcd for C_32_H_46_O_6_Na, 549.3187).

**5**: colorless oil; ${\left[\alpha \right]}_{D}^{20}$ – 20.0 (*c* 0.1, CHCl_3_); IR (ATR) *ν*_max_ 3727, 2961, 2929, 2851, 1734, 1678, 1452, 1367, 1254, 1173, 1027 cm^−1^; ^1^H NMR and ^13^C NMR, see Appendix A; HRESIMS *m*/*z* 551.3310 [M + Na]^+^ (calcd for C_32_H_48_O_6_Na, 551.3343).

**6**: amorphous powder; ${\left[\alpha \right]}_{D}^{20}$ + 45.0 (*c* 0.2, CHCl_3_); IR (ATR) *ν*_max_ 3735, 2971, 2929, 2865, 1724, 1678, 1657, 1452, 1371, 1296, 1173, 1080, 1027 cm^−1^; ^1^H NMR and ^13^C NMR, see Appendix A; HRESIMS *m*/*z* 509.3215 [M + Na]^+^ (calcd for C_30_H_46_O_5_Na, 509.3237).

**7**: colorless oil; ${\left[\alpha \right]}_{D}^{20}$ + 33.3 (*c* 0.1, CHCl_3_); IR (ATR) *ν*_max_ 3735, 2957, 2918, 2848, 1727, 1702, 1657, 1458, 1371, 1254, 1176, 1038 cm^−1^; ^1^H NMR and ^13^C NMR, see Appendix A; HRESIMS *m*/*z* 503.3113 [M + Na]^+^ (calcd for C_31_H_44_O_4_Na, 503.3132).

**8**: colorless oil; ${\left[\alpha \right]}_{D}^{20}$ + 8.3 (*c* 0.2, CHCl_3_); IR (ATR) *ν*_max_ 3727, 2961, 2929, 2851, 1738, 1721, 1671, 1505, 1452, 1374, 1246, 1031 cm^−1^; ^1^H NMR and ^13^C NMR, see Appendix A; HRESIMS *m*/*z* 611.3541 [M + Na]^+^ (calcd for C_34_H_52_O_8_Na, 611.3554).

**9**: colorless oil; ${\left[\alpha \right]}_{D}^{20}$ – 6.7 (*c* 0.1, CHCl_3_); IR (ATR) *ν*_max_ 2961, 2925, 2851, 1745, 1727, 1505, 1265, 1031 cm^−1^; ^1^H NMR and ^13^C NMR, see Appendix A; HRESIMS *m*/*z* 522.3810 [M + NH_4_]^+^ (calcd for C_30_H_52_NO_6_, 522.3789).

**10**: colorless oil; ${\left[\alpha \right]}_{D}^{20}$ + 73.3 (c, 0.1, CHCl_3_); IR (ATR) *ν*_max_ 3477, 3388, 2966, 2923, 2866, 1729, 1457, 1368, 1250 cm^−1^; ^1^H NMR and ^13^C NMR, see Appendix A; HRESIMS *m*/*z* 597.3404 [M + Na]^+^ (calcd for C_33_H_50_O_8_Na, 597.3398).

**11**: colorless oil; ${\left[\alpha \right]}_{D}^{20}$ + 40.0 (*c* 0.1, CHCl_3_); IR (ATR) *ν*_max_ 2965, 2918, 2855, 1731, 1649, 1458, 1374, 1250, 1169, 1035 cm^−1^; ^1^H NMR and ^13^C NMR, see Appendix A; HRESIMS *m*/*z* 597.3396 [M + Na]^+^ (calcd for C_33_H_50_O_8_Na, 597.3398).

**12**: colorless oil; ${\left[\alpha \right]}_{D}^{20}$ + 71.7 (*c* 0.2, CHCl_3_); IR (ATR) *ν*_max_ 3392, 2946, 2925, 2858, 1734, 1455, 1367, 1243, 1180 cm^−1^; ^1^H NMR and ^13^C NMR, see Appendix A; HRESIMS *m*/*z* 595.3241 [M + Na]^+^ (calcd for C_33_H_48_O_8_Na, 595.3241).

**13**: amorphous powder; ${\left[\alpha \right]}_{D}^{20}$ + 6.7 (*c* 0.2, CHCl_3_); IR (ATR) *ν*_max_ 3727, 2957, 2929, 2848, 1727, 1657, 1455, 1374, 1278, 1176 cm^−1^; ^1^H NMR and ^13^C NMR, see Appendix A; HRESIMS *m*/*z* 539.3325 [M + Na]^+^ (calcd for C_31_H_48_O_6_Na, 539.3343).

**14**: colorless oil; ${\left[\alpha \right]}_{D}^{20}$ + 31.7 (*c* 0.2, CHCl_3_); IR (ATR) *ν*_max_ 3717, 2961, 2925, 2872, 1727, 1649, 1458, 1374, 1275, 1257, 1176 cm^−1^; ^1^H NMR and ^13^C NMR, see Appendix A; HRESIMS *m*/*z* 537.3175 [M + Na]^+^ (calcd for C_31_H_46_O_6_Na, 537.3187).

**15**: colorless oil; ${\left[\alpha \right]}_{D}^{20}$ + 30.0 (*c* 0.2, CHCl_3_); IR (ATR) *ν*_max_ 3735, 2957, 2925, 2869, 1731, 1448, 1363, 1278, 1176, 1014 cm^-1^; ^1^H NMR and ^13^C NMR, see Appendix A; HRESIMS *m*/*z* 551.3366 [M + Na]^+^ (calcd for C_32_H_48_O_6_Na, 551.3343).

**16**: colorless oil; ${\left[\alpha \right]}_{D}^{20}$ – 70.0 (*c* 0.2, CHCl_3_); IR (ATR) *ν*_max_ 3727, 2961, 2922, 2851, 1738, 1646, 1452, 1381, 1225, 1021 cm^−1^; ^1^H NMR and ^13^C NMR, see Appendix A; HRESIMS *m*/*z* 467.2762 [M + Na]^+^ (calcd for C_27_H_40_O_5_Na, 467.2768).

## 4. Conclusions

A total of 15 novel scalaranes **1**–**14** and **16**, including 14 bishomoscalaranes and one scalarin derivative, has been isolated from the marine sponge, *Dysidea* sp., and characterized using a combination of 1D and 2D NMR spectroscopy. The isolation and structural identification of compound **15** resulted in the reassignment of the previously characterized 12-*epi*-phyllactone D/E. The actual structure of the reported 12-*epi*-phyllactone D/E was determined to be a mixture of known phyllactones D and E through the precise analysis of the experimental and reported ^13^C chemical shifts. In addition, the effect of the C-18 configuration in **16** on the formation of the hemiacetal E-ring was rationalized by measuring the atomic distances between C-25 and O-24 in **16** and 18-*epi*-**16**. Finally, the evaluation of the anticancer activities of compounds **1**–**16** against MDA-MB-231 revealed that compound **7** exhibited significant cytotoxicity with a GI_50_ value of 4.2 μM. Detailed studies to elucidate the biological mechanism of **7** are currently underway in our laboratory.

## Figures and Tables

**Figure 1 marinedrugs-19-00627-f001:**
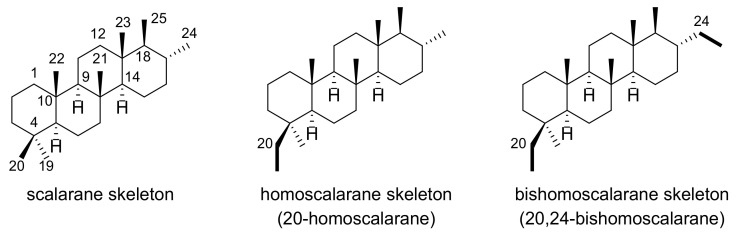
Subtypes of the scalarane skeleton found in marine nature.

**Figure 2 marinedrugs-19-00627-f002:**
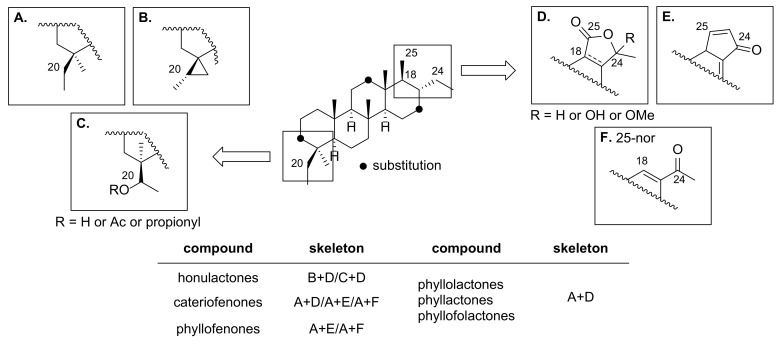
Diversity of bishomoscalarane skeletons frequently found in nature.

**Figure 3 marinedrugs-19-00627-f003:**
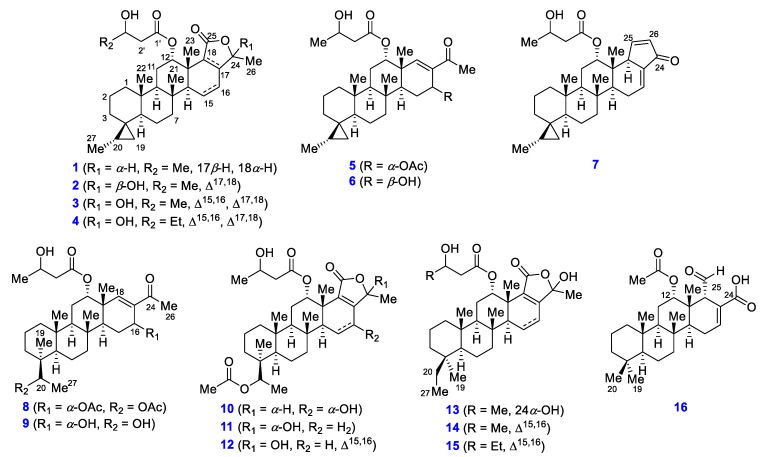
The structures of compounds **1**–**16** isolated from *Dysidea* sp.

**Figure 4 marinedrugs-19-00627-f004:**
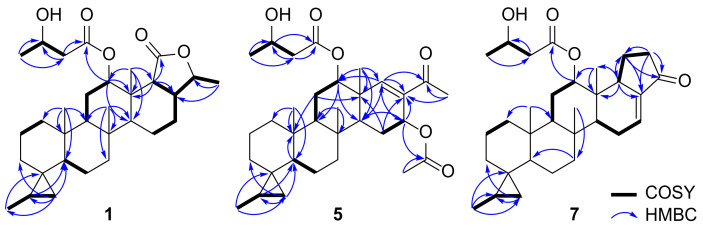
COSY and HMBC correlations observed for compounds **1**, **5**, and **7**.

**Figure 5 marinedrugs-19-00627-f005:**
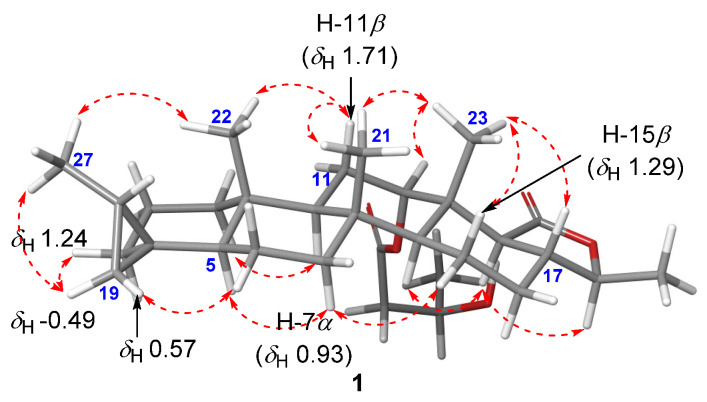
NOESY correlations observed for compound **1**.

**Figure 6 marinedrugs-19-00627-f006:**
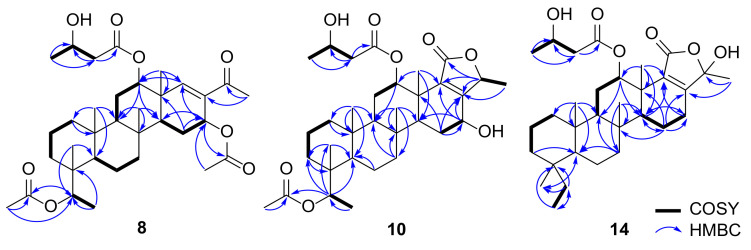
COSY and HMBC correlations observed for compounds **8**, **10**, and **14**.

**Figure 7 marinedrugs-19-00627-f007:**
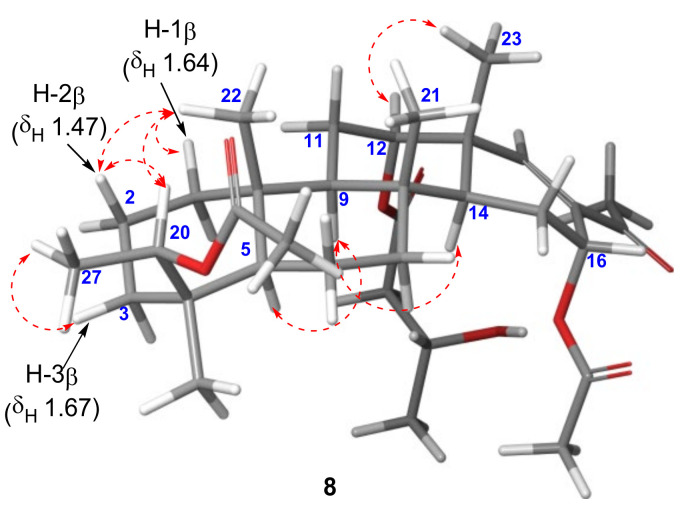
NOESY correlations observed for compound **8**.

**Figure 8 marinedrugs-19-00627-f008:**
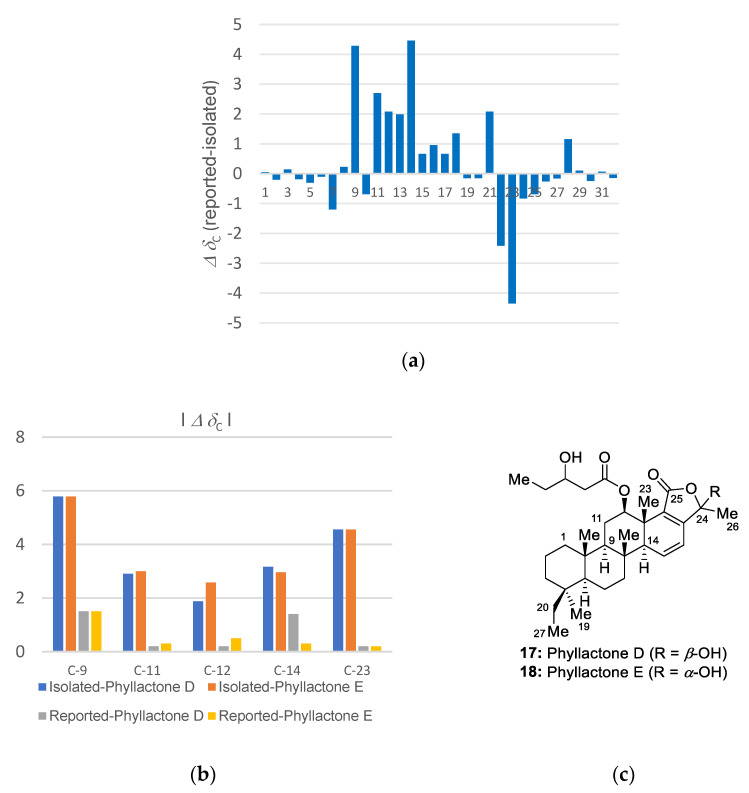
(**a**) The differences observed between the experimental and reported ^13^C chemical shifts (Li et al.) of **15** in CDCl_3_. (**b**) The deviations in the ^13^C NMR chemical shifts observed for reported (Li et al.) and isolated **15** relative to phyllactone D and phyllactone E. (**c**) The structures of phyllactone D and E.

**Figure 9 marinedrugs-19-00627-f009:**
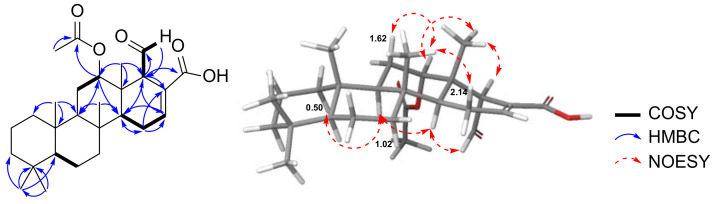
COSY, HMBC, and NOESY correlations observed for compound **16**.

**Figure 10 marinedrugs-19-00627-f010:**
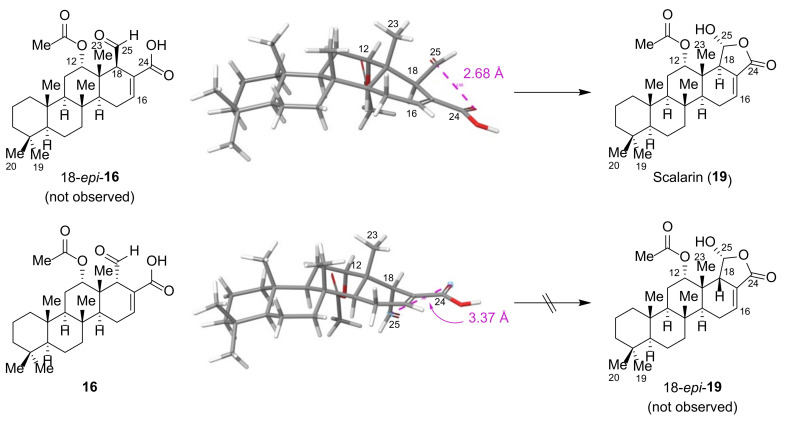
Atomic distances between O-24 and C-25 in compounds **16** and 18-*epi*-**16**.

**Table 1 marinedrugs-19-00627-t001:** The results of the cytotoxicity tests against MDA-MB-231 (human breast cancer cell line) obtained for compounds **1**–**16**.

		Cytotoxicity ^1^ (MDA-MB-231)						
**Compound**	**1**	**2**	**3**	**4**	**5**	**6**	**7**	**8**
GI_50_ (μM)	69.94	43.38	72.49	54.02	53.58	50.8	4.21	53.55
**Compound**	**9**	**10**	**11**	**12**	**13**	**14**	**15**	**16**
GI_50_ (μM)	>100	>100	71.14	>100	50.71	63.54	40.82	>100

^1^ Cisplatin (Sigma-Aldrich, St. Louis, MO) was used as a positive control (GI_50_ = 1.31 μM).

## Data Availability

All data presented in this study are available from the corresponding author on reasonable request.

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
