# Peer review of "Isolation of Scalarane-Type Sesterterpenoids from the Marine Sponge Dysidea sp. and Stereochemical Reassignment of 12-epi-Phyllactone D/E"

_marinedrugs, 2021, doi:10.3390/md19110627_

Round 1

Reviewer 1 Report

This work describes the purification of a total of 30 compounds from the marine sponge from the Philippines. Of the purified compounds, several were new and structural revision for a known compound was also presented. Biological activity assessment was also performed.

  1. Indicate GI50 values for the positive control used
  2. The format of the tabulated NMR data for the compounds, presented in the supplementary information, is difficult to check to verify the correlations presented in the main text and perform a direct comparison with other compounds from the literature. I suggest reformating this and tabulating the 1D and 2D NMR data for each compound.
  3. Indicate the permit for collection/provided scientist for the sample used in this study.

Reviewer 2 Report

Authors claim the identification of 16 new scalarane-type sesterterpenoids. However, compound 15 was reported before by Williams et al as an antitumoral compound (J. Nat. Prod. 2009, 72, 6, 1106–1109).

In the article under review NOESY correlations between H-12 and CH3-23 are used to determine the S configuration of C12, however this NOE interaction is also seen for the 12R* configuration (e.g. Sci Rep 6, 36170 (2016). https://doi.org/10.1038/srep36170). The 3JHH couplings of H12 should be used to establish configuration of C12 unambiguously.

In figure 8a authors compare their observed 13C chemical shifts of compound 5 to what is reported (by reference 11?). However, no differences occur between the chemical shifts observed on compound 15 and those reported by Williams et al (J. Nat. Prod. 2009, 72, 6, 1106–1109). On line 293 it should be stated explicity that Li et al probably analyzed the 12α-epimer.

It is not clear how fig 8b was generated: 'deviations of  chemical shifts observed for reported and isolated 15 from phyllactone D and phyllactone E'=> relative to ?????.

It is obvious that 12α- and β-epimers can be distinguished from their 13C chemical shifts, though to be complete, one should also report the differences in J-couplings of H12. Based on the splitting pattern of this well resolved signal in the proton spectrum, it is possible to distinguish  12β- and α-epimers.

Several spectra in the ‘supplementary files-2’ need phase correction along the directly detected dimension (especially COSY spectra). Labels on the spectra should adjusted according to ‘x-axis assignment/y-axis assignment’.

Values for chemical shifts reported in S1 should all have the same decimal place accuracy.

Round 2

Reviewer 1 Report

Add the permit number and duly acknowledge the partner agencies that provided the permit for collection. This should be included in the acknowledgment section

Reviewer 2 Report

Authors have addressed my remarks adequately. Therefore I believe the manuscript manuscript has been sufficiently improved to warrant publication in Marine Drugs.
